# A Bayesian reanalysis of the effects of hydroxychloroquine and azithromycin on viral carriage in patients with COVID-19

Oliver James Hulme[1,2]*, Eric-Jan Wagenmakers[3], Per Damkier[4], Christopher Fugl Madelung[1,4], Hartwig Roman Siebner[1,5,6], Jannik Helweg-Larsen[7], Quentin F. Gronau[3], Thomas Lars Benfield[6,8], Kristoffer Hougaard Madsen[1,9]

1 Danish Research Centre for Magnetic Resonance, Centre for Functional and Diagnostic Imaging and Research, Copenhagen University Hospital Hvidovre, Hvidovre, Denmark, 2 London Mathematical Laboratory, London, United Kingdom, 3 Department of Psychological Methods, University of Amsterdam, Amsterdam, The Netherlands, 4 Department of Clinical Chemistry & Pharmacology, Odense University Hospital, Odense, Denmark, 5 Department of Neurology, Copenhagen University Hospital Bispebjerg, Copenhagen NV, Denmark, 6 Department of Clinical Medicine, Faculty of Medical and Health Sciences, University of Copenhagen, Copenhagen NV, Denmark, 7 Department of Infectious Diseases, Rigshospitalet, Copenhagen University Hospital, Copenhagen NV, Denmark, 8 Department of Infectious Diseases, Amager Hvidovre Hospital, Hvidovre, Denmark, 9 Department of Applied Mathematics and Computer Science, Technical University of Denmark, Lyngby, Denmark

* oliverh@drcmr.dk

**Data Availability Statement:** All relevant data are within the manuscript and at https://osf.io/5dgmx/.

## Abstract

Gautret and colleagues reported the results of a non-randomised case series which examined the effects of hydroxychloroquine and azithromycin on viral load in the upper respiratory tract of Severe acute respiratory syndrome coronavirus 2 (SARS-CoV-2) patients. The authors reported that hydroxychloroquine (HCQ) had significant virus reducing effects, and that dual treatment of both HCQ and azithromycin further enhanced virus reduction. In light of criticisms regarding how patients were excluded from analyses, we reanalysed the original data to interrogate the main claims of the paper. We applied Bayesian statistics to assess the robustness of the original paper's claims by testing four variants of the data: 1) The original data; 2) Data including patients who deteriorated; 3) Data including patients who deteriorated with exclusion of untested patients in the comparison group; 4) Data that includes patients who deteriorated with the assumption that untested patients were negative. To ask if HCQ monotherapy was effective, we performed an A/B test for a model which assumes a positive effect, compared to a model of no effect. We found that the statistical evidence was highly sensitive to these data variants. Statistical evidence for the positive effect model ranged from strong for the original data ($BF_{+0}$ ~11), to moderate when including patients who deteriorated ($BF_{+0}$ ~4.35), to anecdotal when excluding untested patients ($BF_{+0}$ ~2), and to anecdotal negative evidence if untested patients were assumed positive ($BF_{+0}$ ~0.6). The fact that the patient inclusions and exclusions are not well justified nor adequately reported raises substantial uncertainty about the interpretation of the evidence obtained from the original paper.

**Funding:** The author(s) received no specific funding for this work.

**Competing interests:** Dr. Benfield reports grants from Pfizer, grants from Novo Nordisk Foundation, grants from Simonsen Foundation, grants and personal fees from GSK, grants and personal fees from Pfizer, personal fees from Boehringer Ingelheim, personal fees from Gilead, personal fees from MSD, outside the submitted work; Dr. Siebner reports personal fees from Sanofi Genzyme, personal fees from Novartis, personal fees from Elsevier publishers, other from Springer publishers, outside the submitted work. This does not alter our adherence to PLOS ONE policies on sharing data and materials

## Introduction

In March 2020, at the beginning of the corona virus pandemic, Gautret and colleagues reported the results of a non-randomised open-label case series which examined the effects of HCQ and azithromycin (AZ) on viral load in the upper respiratory tract of SARS-CoV-2 patients. The authors reported that HCQ had virus reducing effects, and that dual treatment of both HCQ and AZ further enhanced virus reduction. At the time, these data triggered urgent speculation whether these drugs should be considered as candidates for the treatment of severe COVID-19. However, questions were quickly raised regarding the study's data integrity, statistical analyses, and experimental design. In light of these questions, we reanalysed the original data by performing a multiverse analysis in which we systematically varied assumptions regarding the inclusion of patients in the analysis, and their clinical status. The Bayesian methods we apply have several advantages in this context. Firstly, in contrast to the frequentist methods applied in the original paper, Bayesian statistics allows for a direct evaluation of the evidence for and against competing hypotheses. In other words, we can go beyond accepting or rejecting the null hypothesis of no treatment effect, by assessing how strong is the evidence provided by the data, for one hypothesis over and another. Secondly, by performing a multiverse analysis in which the data depends on the inclusion assumptions made, we can explore how the strength of evidence for the clinical efficacy of the HCQ treatments depends on which subjects were included or excluded, and what their clinical status was.

## Methods

### Experimental methods

The experimental details are reported in the original published paper [1] which we henceforth refer to as the original paper.

### Treatment groups

For brevity we deviate from the nomenclature of the original paper. $HCQ_{mono}$ refers to treatment with only HCQ. $HCQ_{+AZ}$ refers to treatment with both AZ and HCQ. $HCQ_{group}$ refers to all patients treated with either $HCQ_{mono}$ or $HCQ_{+AZ}$. Comparison group refer to those patients not receiving either treatment. Note that these are erroneously referred to as controls in the original paper. Note that the statistical analysis file available for this reanalysis paper refers to the comparison group as controls.

### Data

The experimental details are reported in the original paper. Raw data was not available at the time of writing but was transcribed from Table 1 of the original paper. We assess the robustness of the original paper's claims by testing four variants of the data, which vary assumptions pertaining to deteriorated and untested patients:

(1) *$Data_{orig}$ is the data as originally reported*. This is the original data, as reported in the original paper.

(2) *$Data_{det}$ includes deteriorated patients*. It is questionable to exclude several patients who could not complete the treatment because their condition deteriorated. This could introduce a selection bias that inflates the effect of the treatment. We therefore modified the original data as follows: of the $HCQ_{mono}$ group, six were originally described as being excluded. One patient died, three deteriorated into intensive care, one patient stopped because of

**Table 1. Descriptive statistics for the study populations.**

| | | HCQ$_{group}$ | | | |
| --- | --- | --- | --- | --- | --- |
| | **Control** | **HCQ$_{mono}$** | **HCQ$_{+AZ}$** | **HCQ$_{group}$ overall** | **Overall** |
| | **(n = 16)** | **(n = 14)** | **(n = 6)** | **(n = 20)** | **(n = 36)** |
| **Age (years)** | | | | | |
| Mean (SD) | 37.4 (24.0) | 52.8 (20.6) | 47.5 (14.1) | 51.2 (18.7) | 45.1 (22.0) |
| Median [Min, Max] | 32.5 [10.0, 75.0] | 51.0 [24.0, 87.0] | 51.5 [20.0, 60.0] | 51.5 [20.0, 87.0] | 47.0 [10.0, 87.0] |
| **Sex** | | | | | |
| Female | 10 (62.5%) | 9 (64.3%) | 2 (33.3%) | 11 (55.0%) | 21 (58.3%) |
| Male | 6 (37.5%) | 5 (35.7%) | 4 (66.7%) | 9 (45.0%) | 15 (41.7%) |
| **Clinical presentation** | | | | | |
| Asymptomatic | 4 (25.0%) | 2 (14.3%) | 0 (0%) | 2 (10.0%) | 6 (16.7%) |
| URTI | 10 (62.5%) | 10 (71.4%) | 2 (33.3%) | 12 (60.0%) | 22 (61.1%) |
| LRTI | 2 (12.5%) | 2 (14.3%) | 4 (66.7%) | 6 (30.0%) | 8 (22.2%) |

This Table summarises the characteristics of the study population. Data was extracted from the Table 1 of Gautret et al. [1]. URTI: Upper respiratory tract infection, LRTI: Lower respiratory tract infection.

nausea and one left the hospital. These can be considered counterfactual cases that are necessary to entertain for a conservative estimate of the effects of HCQ$_{mono}$. In the following, we add the patients who died or entered intensive care to the positive test cases for day 6. This is tabulated in Fig 1A. We exclude both the patients who stopped treatment due to nausea, and the one patient who left the hospital, due to the ambiguity of their cases. This means that four cases are added to the HCQ$_{mono}$ that tested positive for SARS-CoV-2.

(3) *Data$_{xcon}$ includes deteriorated patients and excludes the untested patients*. On day 6 there were 5 patients who were untested, even though the day 6 test outcome was the primary outcome. The untested patients were assumed to be positive in the original paper, and for this data variant we simply exclude them. This can be motivated by the fact that the tests have some level of stochasticity, as can be seen from the fact that there are a total of 9 transitions from negative tests on a given day, followed by positive tests the next day. For 3 of the 5 patients, they tested positive on day 5, and for 2 of the patients the tests were not performed either on day 4 or day 5. Hence it is not known with any certainty what the test outcomes would have been in the five untested patients had they been tested on day 6. This is especially problematic since all 5 of these untested patients belonged to the comparison group. For this reason it is important to analyse data that excludes these untested patients.

(4) *Data$_{negcon}$ includes the deteriorated patients and assumes untested patients test negative*. Given the problem with untested patients, we perform an analysis to evaluate what would happen to the results had these patients been tested and they were negative, rather than positive as assumed in the original data. This is included as a data variant not because it is the most likely case, but because it is the most conservative possible outcome, given the uncertainty of the reported data.

## Statistical analysis

Bayesian statistical analyses of the data were performed in JASP (version 0.11, jasp-stats.org). We note that caution should be taken with reanalyses of this data set because there are some discrepancies between different pre-prints and published versions. The analysis file, including

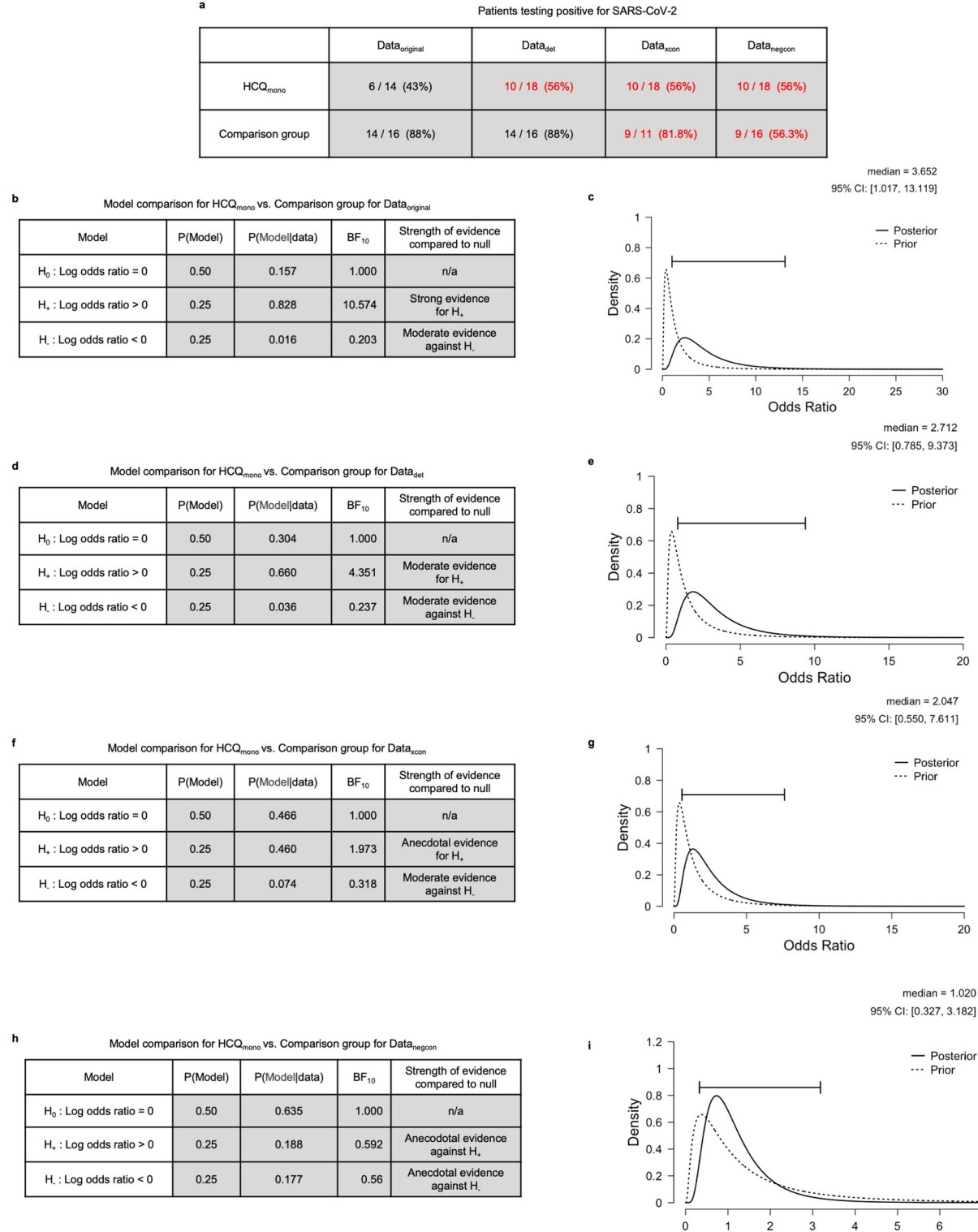

**Fig 1. Main effect of HCQ on SARS-CoV-2 viral carriage reduction. a,** Table shows frequencies and proportions of patients testing positive for SARS-CoV-2 on day 6, as reported in the original paper, here presented under the four data variants. Numbers in red indicate the data that was modified by changing the assumptions, numbers in brackets indicate percentage proportions. **b**, Model comparison table for the main effect of $HCQ_{mono}$ versus comparison group for $Data_{original}$. **c,** Posterior distribution of the odds ratio for $HCQ_{mono}$ compared to comparison group under $Data_{original}$. **d-i** are equivalent plots for the remaining data variants.

the raw data we transcribed from the original paper, and all materials, are available on the Open Science Framework at https://osf.io/5dgmx/. All references to strength of evidence refer to standard conventions for the evidentiary support of Bayes factors (BF) such that 1–3 is classed as anecdotal, 3–10 as moderate, 10–30 as strong, and 30–100 as very strong [2]. For Bayes factors below 1, the reciprocal can be taken to obtain the strength of evidence in the opposite direction. Our initial analyses attempted to reproduce the findings of the original paper using the same data. We then performed the same analyses again, but with modified assumptions for how to treat excluded patients and untested patients. This can be considered a form of sensitivity analysis, of the sort recommended by a statistical review of the original paper [3]. Unless otherwise stated, we focus on the primary outcome of viral carriage on day 6 relative to inclusion point into the study.

## Results

### Main effect of $HCQ_{mono}$ on viral carriage reduction

The original paper compared $HCQ_{group}$, which is a composite of two groups ($HCQ_{mono}$ and $HCQ_{+AZ}$) with different drug treatments, to the comparison group. A more appropriate test for this question would be $HCQ_{mono}$ versus comparison group. Here we perform the test $HCQ_{mono}$ versus comparison group, and assess its sensitivity to the variants of the data under different assumptions regarding deteriorated and untested patients. Fig 1A shows the number and proportion of patients testing positive for SARS-CoV-2 grouped by $HCQ_{mono}$ or comparison group.

Here we quantify the degree to which $HCQ_{mono}$ reduces viral carriage of SARS-CoV-2 (viral carriage hence). We conducted a Bayesian A/B test [4, 5] that considered three rival models. The first model is a null model $H_0$ which states that the viral carriage in $HCQ_{mono}$ is equal to that of the comparison group. This entails that the log odds ratio $\psi$, for viral carriage reduction is equal to 0. The second model is a positive effect model $H_+$ which predicts that the effect of $HCQ_{mono}$ exceeds that of the comparison group, and is thus indicative of a beneficial effect of $HCQ_{mono}$ on viral carriage. Under this model, $\psi$ is assigned a positive-only truncated normal prior distribution $N_+ (\mu,\sigma)$. The third model is a negative effect model $H_-$ that predicts that the effect of $HCQ_{mono}$ is smaller than that of the comparison group, which would indicate a harmful effect of $HCQ_{mono}$ on viral carriage. Under this model $\psi$ is assigned a negative-only truncated normal prior distribution $N_-(\mu,\sigma)$. For all models in this paper, we perform a default analysis in which the parameters of the normal distribution are set such that $\mu = 0$ and $\sigma = 1$. The results are tabulated in Fig 1B, 1D, 1F and 1H for each of the four data variants. The Bayes factors that we report indicate how likely the data is under each model. Thus for each data variant one can use these factors to find which model finds most support from the data. For all data variants, the prior probabilities of each model were the same, namely that $H_0$ is assigned a probability of 0.5, $H_+$ and $H_-$ are each assigned a probability of 0.25. Given these priors, for each of the three models, the corresponding posterior model probabilities are computed P (Model | data) as can be seen in the tables of Fig 1B, 1D, 1F and 1H. For all data variants (except $Data_{negcon}$) it is the positive effect model $H_+$ that receives most support from the data. For $Data_{original}$ the evidence is strong ($BF_{+0} = 10.57$) meaning that the data is approximately 11 times more likely under $H_+$ than under $H_0$. For $Data_{det}$ the evidence is moderate when including the deteriorated patients ($BF_{+0} \sim 4.35$), and for $Data_{xcon}$ the evidence is anecdotal when excluding untested patients ($BF_{+0} \sim 2$). For $Data_{negcon}$ there was anecdotal evidence against the positive effect model if untested patients were assumed negative ($BF_{+0} \sim 0.6$). As is evident from this, the strength of the evidence for the positive effect of $HCQ_{mono}$ over the comparison group is highly sensitive to the assumptions regarding what to do with the deteriorated or

untested patients. The more conservative the assumptions that were made, the lower the strength of the evidence for the viral reduction effect of the HCQ treatment. In other words, different choices in the pre-processing of the data can sway the evidence from strong evidence for a positive effect of $HCQ_{mono}$ to anecdotal evidence against such an effect.

This analysis provides interval estimates that were missing from the original report, which allow us to assess the size of the odds ratios, and their plausible ranges under different assumptions about the data. This sensitivity to assumptions is expressed in the credibility intervals for the odds ratios of the treatments. For $Data_{original}$ the 95% credibility interval for the odds ratio has a lower bound of ~1.02 and an upper bound of ~13. For the $Data_{negcon}$ however the same intervals run from a lower bound of ~0.33 to an upper bound of ~3.2.

## Main effect of combined treatment of AZ and HCQ on viral carriage reduction

Fig 2A shows the number and proportion of patients testing positive for SARS-CoV-2 for each of the HCQ treatment subgroups, $HCQ_{mono}$ and $HCQ_{+AZ}$. Here we quantify the degree to which combining AZ with HCQ reduces viral carriage, over and above the effects of HCQ on

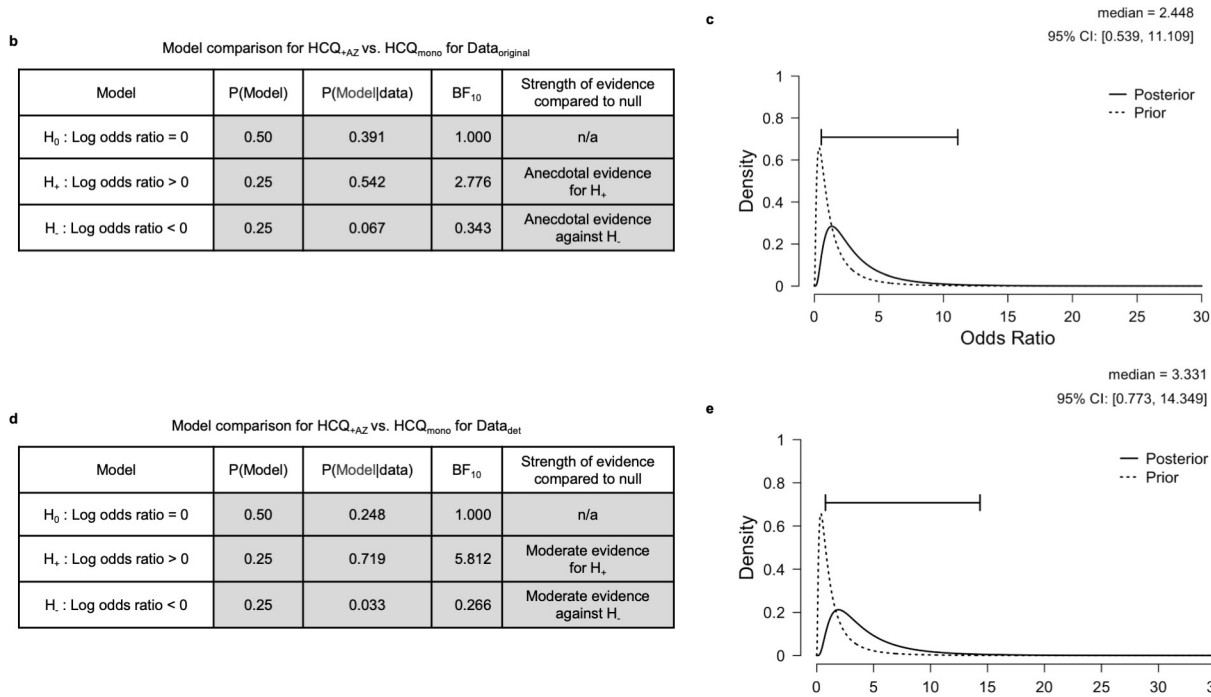

**Fig 2. Main effect of combined treatment of AZ and HCQ on viral carriage reduction. a,** Table shows frequencies and proportions of patients testing positive for SAR-COV-2 on day 6, for both $Data_{original}$ and $Data_{det}$. The other two data variants are not included since they are identical to $Data_{det}$. Numbers in red indicate the data that was modified by changing the assumptions, numbers in brackets indicate equivalent rounded percentages. **b,** Model comparison table for the main effect of $HCQ_{+AZ}$ versus $HCQ_{mono}$ for $Data_{original}$. **c,** Posterior distribution of the odds ratio for $HCQ_{+AZ}$ versus $HCQ_{mono}$ for $Data_{original}$ **d** and **e** are equivalent plots for $Data_{det}$.

its own. We again conducted an A/B test that considered three rival models. The first model is a null model $H_0$ which states that the viral carriage in $HCQ_{mono}$ is equal to that of the $HCQ_{+AZ}$, thus offering no clinical benefit or harm, in terms of viral carriage. This entails that the log odds ratio $\psi$, for viral carriage reduction is equal to 0. The second model is a positive effect model $H_+$ which predicts that the effect of $HCQ_{+AZ}$ exceeds that of the $HCQ_{mono}$, and is thus indicative of a beneficial effect of adding AZ to HCQ to reduce viral carriage. Under this model $\psi$ is assigned a positive-only normal prior distribution $N_+(\mu,\sigma)$. The third model is a negative effect model $H_-$ that predicts that the effect of $HCQ_{+AZ}$ is smaller than that of $HCQ_{mono}$, which would indicate a harmful effect of adding AZ to HCQ in terms of viral carriage. Under this model $\psi$ is assigned a negative-only normal prior distribution $N_-(\mu,\sigma)$.

The results are tabulated in Fig 2 for each of the two data variants, $Data_{original}$ and $Data_{det}$. As with the previous model, for both data variants, the prior probabilities of each model were the same, namely that $H_0$ is assigned a probability of 0.5, $H_+$ and $H_-$ are each assigned a probability of 0.25. Given these priors, for each of the three models the corresponding posterior model probabilities are computed P(Model | data), as can be seen in the tables of Fig 2B and 2D. For both data variants it is the positive effect model $H_+$ that recieves most support from the data. Note that only two data variants are computed since the comparison group is not part of this test.

For $Data_{original}$ the evidence is anecdotal ($BF_{+0} = 2.776$) meaning that the data is approximately 3 times more likely under $H_+$ than under $H_0$. This level of evidence is sometimes referred to as "barely worth mentioning" [2]. This would appear to temper the conclusions of the original paper, which inferred that this was a clinically important result, and one that was central to the medical recommendations of the paper. Given the priors described above, the corresponding posterior model probabilities would be $H_0$ (0.391), $H_+$ (0.542) and $H_-$ (0.067). As can be seen in Fig 2C the 95% credibility interval for the odds ratio has a lower bound of ~0.54 and an upper bound of ~11. This indicates that there is also large uncertainty in the size of the positive clinical effect. The lower bound represents a 46% reduced chance of viral clearance having been improved by adding AZ to HCQ. The upper bound of this estimate represents a 1000% improved chance. The large uncertainty in this estimate of the odds ratio is due to the small sample size obtained in the original findings, in which the $HCQ_{+AZ}$ group had only 6 members.

For $Data_{det}$ the evidence is moderate when including the deteriorated patients ($BF_{+0}$ ~5.812), meaning that the data is approximately 6 times more likely under $H_+$ than under $H_0$. This demonstrates that the more conservative exclusion criteria actually increases the strength of evidence for the superiority of $HCQ_{+AZ}$ over $HCQ_{mono}$. This is because including the deteriorated patients negatively impacts on the proportion of negative tests for the $HCQ_{mono}$ group but not the $HCQ_{+AZ}$ group. If the null model $H_0$ were assigned prior probability 0.5 and the $H_+$ and $H_-$ were each assigned a probability of 0.25, the corresponding posterior model probabilities would be $H_0$ (0.248), $H_+$ (0.719) and $H_-$ (0.033). As shown in Fig 2E the 95% credibility interval for the odds ratio has a lower bound of ~0.77 and an upper bound of ~14, indicating a large uncertainty in the positive clinical effect. The lower bound represents a 23% reduced chance of viral clearance having been improved by adding AZ to HCQ. The upper bound of this estimate represents a 1300% improved chance. Again, the large uncertainty in this estimate is due to the small sample size obtained in the original findings, in which the $HCQ_{+AZ}$ group had only 6 members.

As is evident from this analysis, the strength of the evidence for the positive effect of $HCQ_{+AZ}$ over $HCQ_{mono}$ is sensitive to the assumptions regarding what to do with the deteriorated patients. Different choices change the evidence, from anecdotal based on the original data, to moderate under more conservative exclusion criteria. This analysis provides interval

estimates that were missing from the original report, which importantly allow assessment of the odds ratios, and their plausible ranges under different assumptions about the data. For both data variants there is large uncertainty in the odds ratios, ranging from moderate reductions in the chance of improvement, up to very large chances of improvement.

The strength of evidence for all statistical comparisons for all data variants is shown in Fig 3. Note that we have focused on the comparisons of $HCQ_{mono}$ versus comparison group, and $HCQ_{+AZ}$ vs. $HCQ_{mono}$, because these answer the questions set out in the original paper. We computed two other comparisons for completeness, $HCQ_{group}$ versus the comparison group, and $HCQ_{+AZ}$ versus the comparison group. Focus on these last two tests were downgraded, as because the first test aggregates two different treatments, and the second test confounds the effect of AZ. The full analysis details are available in the supplementary materials. As can be seen for both of these additional tests, the evidence is sensitive to the assumptions pertaining to the inclusion of deteriorated patients as well as to the status of untested patients.

## Discussion

### Summary

Using a complementary (Bayesian) statistical framework, we evaluated the strength of the statistical evidence for the main claims of Gautret et al., and we asked how robust this evidence is to different assumptions about how to treat deteriorated and untested patients. Though we were able to qualitatively reproduce a positive effect of HCQ on viral load reduction, and a further improvement by adding AZ, the strength of the evidence was highly sensitive to variations in these assumptions. We discussed these in detail and provided a broader context for evaluating the quality of evidence offered by the original paper. In the original paper, the main test for the effect of HCQ was performed by comparing a group of two different treatments (monotherapy $HCQ_{mono}$ and the combination therapy $HCQ_{+AZ}$) against the comparison group. This test does not directly answer the question of what is the clinical effect of HCQ on upper respiratory tract SARS-CoV-2 viral load reduction, and to answer this one needs to compute the effect of $HCQ_{mono}$ against the comparison group. It is regrettable that this test was not reported because it is the test that is necessary to evaluate the effect of HCQ on viral reduction. Performing a Bayesian A/B test, we found that for the original data, there was strong statistical evidence for the positive effect of $HCQ_{mono}$ improving the chances of viral reduction when compared to the comparison group. However, we found that the level of evidence drops down to moderate evidence when including the deteriorated patients, and it drops further to

Strength of evidence for $H_+$ over $H_0$ ($BF_{+0}$)

| Comparison between $H_+$ vs. $H_0$ | $Data_{original}$ | $Data_{det}$ | $Data_{xcon}$ | $Data_{negcon}$ |
|---|---|---|---|---|
| $HCQ_{mono}$ vs. Comp. group | Strong 10.57 | Moderate 4.35 | Anecdotal 1.973 | Anecdotal 0.59 |
| $HCQ_{+AZ}$ vs. Comp. group | Very strong 65.80 | Very strong 65.80 | Strong 21.31 | Moderate 5.50 |
| $HCQ_{+AZ}$ vs. $HCQ_{mono}$ | Anecdotal 2.78 | Moderate 5.81 | Moderate 5.81 | Moderate 5.81 |
| $HCQ_{group}$ vs. Comp. group | Very strong 82.92 | Strong 23.06 | Moderate 6.97 | Anecdotal 1.16 |

**Fig 3. Summary of strength of evidence for positive effect model over null model.** This table summarises all the Bayesian A/B tests performed, their resulting Bayes factors, and their associated descriptions of evidence strength. As per our notation throughout, $H_+$ is the positive effect model and $H_0$ is the null model of no effect. The left column indicates the patient groups to which these models are applied. The central and right columns can be read as quantifying the strength of evidence for a beneficial effect on viral carriage. The Bayes factors $BF_{+0}$ are shown in grey, indicating how many times more likely the data is under $H_+$ than under $H_0$. Red text indicates evidence against $H_+$.

anecdotal evidence when excluding the patients that were not tested on the day of the primary outcome (day 6). For context, anecdotal evidence is generally considered 'barely worth mentioning' [2]. We were able to qualitatively reproduce the finding of an improvement of $HCQ_{+AZ}$ over $HCQ_{momo}$. However, although this finding was statistically significant in the original finding, our reanalysis revealed only anecdotal evidence for the positive effect of $HCQ_{+AZ}$ over $HCQ_{momo}$. However, when we included the deteriorated patients into the analysis, this evidence increased to moderate. We also performed another test, which is to compare $HCQ_{+AZ}$ against the comparison group. It should be noted that this test is not the most relevant test because it varies two drugs at the same time. Nevertheless, the statistical evidence for the positive effect of the combined treatment over no treatment is very strong for the original data, and drops down to moderate when excluding the untested patients (Fig 3).

## Statistical considerations

Common to both of these sets of analyses is the fact that they are highly sensitive to the assumptions that are made about the exclusion of patients, and of the test status of untested patients. That these assumptions are not well justified, nor adequately reported gives reason to be cautious in the interpretation of the evidence obtained from the original paper. Furthermore, this uncertainty is generally reflected in the credibility interval estimates for the odds ratios, which can range from moderate decreases or small increases in the chances of improvement, through to very large chances of improvement. These uncertainties stem from the small sample sizes for each subgroup. Indeed, the original paper has been criticised for being underpowered due to its relatively small sample size (16 comparison group, 20 treated). The Bayesian inference framework is a natural way to incorporate sample size in the analysis. Small sample sizes will lead to more variance and hence less certainty on the estimated model parameters, and as such typically do not provide compelling evidence; as sample size increases, the evidence will generally become stronger, either in favor of H0 (when there is no discernable effect in the data) or in favor of H1 (when the data do show an effect). Thus, as sample size grows the evidence typically grows as well, in a smooth fashion. We argue that, from the perspective of statistical evidence, this criticism of being underpowered is less relevant once the data are observed. Firstly, such criticisms, though commonly espoused, should be made with reference to the effect size they are underpowered for. Small sample studies can be well powered for detecting large effect sizes. More importantly, although estimating power is useful in planning experiments, it can be misleading when making inferences from observed data [6]. In this reanalysis we rely on Bayes factors, which are an extension of likelihood ratios beyond point hypotheses. These methods of inference do not average over hypothetical replications of an experiment, but instead condition on the data that were actually observed. For instance, the fact that a small sample can reveal strong evidence for an effect indicates that the effect size could be relatively large. In this way, Bayes factors rationally quantify the evidence that a particular dataset provides for or against the null, or any other hypothesis. Recourse to claims about the power of an experiment can be displaced by considering the strength of the evidence for one model over other models. This is clinically important because the strength of this evidence is not apparent from the statistical reporting of the original paper (which only reported p-values). Put simply, the findings of the original paper cannot be dismissed solely on the basis of being "underpowered".

## Negative effect hypotheses

In this paper we assigned prior probabilities of 0.5 to $H_0$ and 0.25 each to $H_+$ and $H_-$. We assign probability mass to both the negative and positive hypotheses because when testing a drug it is

important to know whether it negatively or positively impacts on clinical outcomes. A different model which only considered only $H_0$ and $H_+$ which, say, assigned prior probability of 0.5 to both hypotheses would have assigned higher posterior probabilities to the $H_+$ than those reported here. However, these different priors on the hypotheses would not change any of the Bayes factors reported, because such priors cannot influence the Bayes factor. This is one of the reasons we have emphasised the interpretation of the Bayes factor rather than the posterior probabilities of the hypotheses.

## Experimental design and pre-registered protocol

The most fundamental problem with the original paper is that there was no randomisation of the treatment, which means it is vulnerable to differences in baseline risk between the sub-groups. In the original paper, the treatment groups are confounded by several variables including whether or not they met the exclusion criteria, which centre implemented treatment, and differences in consent (the comparison group were composed of those that met the exclusion criteria or did not consent to treatment). For a full statistical review of these considerations see Dahler et al [3]. Most importantly, the comparison between $HCQ_{+AZ}$ and $HCQ_{mono}$ is confounded by the unreported clinical reasons for which the physicians decided to add the AZ treatment to some patients but not to others. If these reasons were important enough to warrant different treatment, then they are important enough to impact on the comparability between the two groups. Whilst we refrain from making formal inferences, it is relevant to note that the $HCQ_{group}$ patients were older than the comparison group patients (median age 51.5 and 32.5 years respectively, Table 1). It is also worth mentioning, that the comparison group included five cases aged 16 or younger, which should again warrant caution when comparing outcomes between groups. We briefly comment on the existence of putative deviations from the pre-registered protocol, available at https://www.clinicaltrialsregister.eu/ctr-search/trial/2020-000890-25/FR. Outcomes specified in advance included evaluation of upper respiratory tract viral carriage at 1,7, and 14 days, and yet the primary outcome reported in the paper was on day 6. This has been interpreted by some as outcome switching, however we would, in the absence of further information, suggest the possibility that this is an issue of how the days are numbered, whether one starts counting from zero or one. The day 14 outcome was presumably not included such that the report could be published 7 days earlier, which is defensible given the urgency of the pandemic at the time of writing. The secondary outcomes registered in the protocol are not adequately reported or analysed. Finally, the raw data tables changed between different versions of the preprint and the published paper, and thus questions can be asked about data integrity. Clearly, accommodation must be made for the speed at which the original report was published, and the conditions under which the data were presumably collected. The integrity of the reanalysis presented here is explicitly predicated on the assumption that all these possible deviations and data integrity issues can be adequately resolved. Good clinical practice inspection for the sake of patient safety and data transparency would help to resolve such issues.

## Measurement of viral load

It is important to note that the PCR based test uses a threshold of 35 cycles (CT) to distinguish between PCR positive and PCR negative, some PCR positive patients in particular in the HCQ treatment group show CT numbers that are quite close to this threshold indicating that the status might be somewhat ambiguous during the test. Furthermore, a number of patients are later tested positive after being tested negative (occurring a total 9 times in 8 patients) which may further question the use of a hard threshold on the number of cycles. For these reasons using

duplicate sample analysis and confirmatory tests and eventually developing quantitative PCR tests for assessment of treatment effects would be recommended for future studies. Also note, that the current recommendation for a FDA-emergency approved test is that negative PCR results do not preclude presence of SARS-CoV-2 infection and recommend that such results be accompanied by clinical observations, patient history, and epidemiological information. Finally, it is important to determine whether SARS-CoV-2 virus nucleic acid detected by PCR is replication competent or not. At the time of writing, detailed clinical outcome data was not available, precluding any analysis relevant to a clinical outcome other than change from a positive to a negative PCR-based test.

## Clinical safety

While the viral load measurements were noisy, showing multiple reversals between test outcomes, there is greater certainty around other clinical outcomes such as the 4 patients whose condition seriously deteriorated. It is important to stress that all of these belonged to the $HCQ_{mono}$ group, a fact that did not adequately temper the central claims of the original paper regarding the clinical potential of HCQ. Another way to state this would be that, though there is varying degrees of evidence for an effect of HCQ on viral load, it is known with greater certainty that all of the deteriorations occurred in the HCQ treatment group. Greater weight should be placed on this fact, when stating the possible clinical benefits of HCQ in the treatment of Covid-19.

## Conclusions

We find that computing the appropriate statistical tests for the effect of HCQ on viral load reduction, yields results that are highly sensitive to the assumptions about which patients are included and how. While this evidence is strong for the assumptions made by the original paper, for more conservative assumptions, the evidence is substantially weaker than originally reported. Performing the same analysis approach to the question of whether AZ improves HCQ treatment, we find moderate statistical evidence for a positive effect. Whether this is a meaningful comparison however is questionable, based on the fact that it is confounded by undisclosed clinical decision making, that lead to some being treated with AZ and others not. To be clear, our analysis does not resolve the uncertainties that follow from the original experimental design, nor does it address concerns that have been raised about the study's data integrity. The only way to resolve these will be via the randomised controlled trials (RCT) that are already underway.

## Acknowledgments

Due to the rapid pace of this analysis we solicited the assistance of a large number of informal reviewers. Many of these we cannot identify simply because they were anonymised by the collaborative text editing software. They know who they are, and we thank them. Those whose identities we known are Elisabeth Bik, Andrew A Love, Gaetan Burgio, and Birgitte Madsen, and we thank them too.

## Author Contributions

**Conceptualization:** Oliver James Hulme, Per Damkier, Kristoffer Hougaard Madsen.

**Data curation:** Oliver James Hulme, Christopher Fugl Madelung.

**Formal analysis:** Oliver James Hulme, Eric-Jan Wagenmakers, Quentin F. Gronau, Kristoffer Hougaard Madsen.

**Methodology:** Oliver James Hulme, Eric-Jan Wagenmakers, Per Damkier, Quentin F. Gronau, Thomas Lars Benfield.

**Project administration:** Oliver James Hulme.

**Supervision:** Hartwig Roman Siebner, Thomas Lars Benfield.

**Visualization:** Oliver James Hulme.

**Writing – original draft:** Oliver James Hulme.

**Writing – review & editing:** Oliver James Hulme, Eric-Jan Wagenmakers, Per Damkier, Christopher Fugl Madelung, Hartwig Roman Siebner, Jannik Helweg-Larsen, Quentin F. Gronau, Thomas Lars Benfield, Kristoffer Hougaard Madsen.

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
