## [Decision Letter · Decision Letter 0]

18 Jun 2020

PONE-D-20-11807

A Bayesian reanalysis of the effects of hydroxychloroquine and azithromycin on viral carriage in patients with COVID-19

PLOS ONE

Dear Dr. Hulme,

Thank you for submitting your manuscript to PLOS ONE. After careful consideration, we feel that it has merit but does not fully meet PLOS ONE’s publication criteria as it currently stands. Therefore, we invite you to submit a revised version of the manuscript that addresses the points raised during the review process.

As you will see, reviewers made comments that should be very simple for you to address. In particular, I would take seriously the suggestions to improve the ms in terms of its didactical value. On a more personal note, I'd like to thank you for performing this analysis. It is quite timely, given the recent fuss about hydroxychloroquine and connected Lancet study :)

We look forward to receiving your revised manuscript.

Kind regards,

Jean Daunizeau

Academic Editor

PLOS ONE

Journal Requirements:

2) We note that you have stated that you will provide repository information for your data at acceptance. Should your manuscript be accepted for publication, we will hold it until you provide the relevant accession numbers or DOIs necessary to access your data. If you wish to make changes to your Data Availability statement, please describe these changes in your cover letter and we will update your Data Availability statement to reflect the information you provide.

3) Thank you for stating the following in the Competing Interests section:

[Dr. Benfield reports grants from Pfizer, grants from Novo Nordisk Foundation, grants

from Simonsen Foundation, grants and personal fees from GSK, grants and personal fees from Pfizer, personal fees from Boehringer Ingelheim, personal fees from Gilead,

personal fees from MSD, outside the submitted work; Dr. Siebner reports personal fees

from Sanofi Genzyme, personal fees from Novartis, personal fees from Elsevier

publishers, other from Springer publishers, outside the submitted work].

4) Please ensure that you refer to Figure 4 in your text as, if accepted, production will need this reference to link the reader to the figure.

5) Please include a copy of Table 4 which you refer to in your text.

Reviewers' comments:

Reviewer's Responses to Questions

**Comments to the Author**

1. Is the manuscript technically sound, and do the data support the conclusions?

Reviewer #1: Yes

Reviewer #2: Yes

2. Has the statistical analysis been performed appropriately and rigorously? 

Reviewer #1: Yes

Reviewer #2: Yes

3. Have the authors made all data underlying the findings in their manuscript fully available?

Reviewer #1: Yes

Reviewer #2: Yes

4. Is the manuscript presented in an intelligible fashion and written in standard English?

Reviewer #1: Yes

Reviewer #2: Yes

5. Review Comments to the Author

Reviewer #1: PONE-D-20-11807

This manuscript by Hulme et al. is a data reanalysis study and reply to Gautret et al. 20201.

It is entitled: “Reply yo Gautret et al. 2002: A Bayesian reanalysis of the effects of hydroxychloroquine and azythromicin on viral carriage in patients with COVID-19”.

General context: the original article by Gautret and collaborators came out around mid-March, that is to say very quickly and in the early phase of the COVID-19 epidemic in Europe. It concluded for strong evidence in favour of using hydroxychloroquine to treat patients, and for even stronger evidence for combining hydroxychloroquine with azythromicin. As one of the first clinical study, following in vitro experiments, this paper and its conclusions have been widely used as a support to promote the use of hydroxychloroquine to treat COVID-19 patients. A promotion which, given the urgency of fighting the pandemic, had an influence far beyond the scientific community alone, on public debate, political actions and individual behavior, in several countries. However, this study has been heavily criticized for several aspects of its design which clearly deviates from the good scientific practices prescribed for clinical trials. Most of those criticisms are reminded by the authors here, who also point to an early statistical review of the original paper by Dahly et al.2. The main criticisms in this review include the lack of randomization, the lack of a covariate adjusted analysis (such as age, gender, degree of illness, medical center…), unethical practice by the inappropriate inclusion of control patients who refused the treatment, and the case of patients who were inappropriately dropped from the analysis. As pointed out by the authors of the current manuscript, all of these weaknesses, not to say violations, generate uncertainty as to the findings of the original study, which contrasts with the overestimated positive conclusion of its authors.

Aim of the current study: Hulme and colleagues propose a reanalysis of the above data, in order to draw a more rigorous and precise conclusion, with estimating the associated uncertainty. Therefore, and contrary to the original paper, they use Bayesian A/B tests which enable to quantify the evidence in favour of a given hypothesis. The authors focus on two questions: (i) the main effect of hydroxychloroquine on viral carriage, and (ii) the main effect of combined azythormicin and hydroxychloroquine on viral carriage. The outcome is the fraction of patients in each group (hydroxychloroquine, hydroxychloroquine+azythromicin, comparison group) with a positive test at day 6. Importantly, the main added-value of this study is to quantify the effect of the uncertainty around some key aspects of the data, namely the exclusion of some patients who deteriorated and could not be tested at day 6, and missing tests for some patients in the comparison group. These missing data were discarded in the original study, despite the fact that their influence could be significant given the small samples in each group. In the current study, the authors consider different scenarios with these data, namely conservative ones. They also perform more appropriate tests given the questions of interest. They conclude for substantially weaker evidence than initially reported.

Main comments: This paper is important, timely and fairly well written. It brings up an important twofold message to be reminded: (i) despite the emergency of the situation, all efforts should be made to acquire relevant data in order to be able to draw any reliable conclusion; (ii) given imperfect data, the most appropriate and careful statistical analysis should be performed to draw fair conclusions.

The current paper could be better organized in order to be clearer and more straightforward, focusing on its main message. I recommend the authors:

• To introduce the data early in the paper (groups, sample sizes, specify to which group the untested patients and the ones who deteriorated belonged to, …). Hence Figure 4 should become figure 1.

• To shorten the abstract and make it more impactful.

• To move all the information in the Results section that pertain to a methodological description, to the Methods section.

Regarding the statistical approach that is at the heart of this study, I have the following questions:

• The authors of the original paper could argue that what motivated them to use the hydroxychloroquine is a bundle of objective arguments (the known effect of this molecule, the outcome of recent in vitro tests…). Hence a more appropriate test may only consider two hypothesis, H0 and H+, with 0.5 prior probability for each of them. I recommend the authors to discuss this point of view and eventually report the outcome of the tests based on such prior hypothesis.

• The authors perform several tests on the same data. To what extend the Bayesian approach they used should consider correcting for multiple testing? (see e.g. 3).

• It could be useful to also report the exceedance probability for H+, a quantity that directly pertain to the question of interest, that is how more probable is my hypothesis of interest compared to alternative ones.

• In the discussion, the authors develop an interesting explanation around the notion and impact of an underpowered study. For educational reason at least, it would be interesting to explain the influence of the number of samples here and how the Bayesian approach accounts for it compared to a classical, frequentists approach as used in the original paper.

Minor remarks: I noticed the following typos

- In Methods / Data / (2): “We exclude both the patient WHO …”

- In Results / Main effect of HCQ… / 3rd paragraph: “…untested patients were assumed NEGATIVE”

Conclusion: I do recommend this paper to be published in PLoS ONE, after the authors would have addressed the above comments.

References

1. Gautret, P. et al. Hydroxychloroquine and azithromycin as a treatment of COVID-19: results of an open-label non-randomized clinical trial. International Journal of Antimicrobial Agents 105949 (2020) doi:10.1016/j.ijantimicag.2020.105949.

2. Dahly, D., Gates, S. & Morris, T. Statistical review of Hydroxychloroquine and azithromycin as a treatment of COVID-19: results of an open-label non-randomized clinical trial. https://zenodo.org/record/3724167 (2020) doi:10.5281/ZENODO.3724167.

3. Sjölander, A. & Vansteelandt, S. Frequentist versus Bayesian approaches to multiple testing. Eur J Epidemiol 34, 809–821 (2019).

Reviewer #2: This paper provides a careful re-analysis of Gautret et al. data using Bayesian statistics that these authors presumably do not know (or understand if I may say). The analysis sounds solid and conclusions are fairly presented.

I appreciate the need for the re-analysis of the data of Gautret et al. but it is unclear what the scientific message of the paper is.

Given that the Gautret et al. study is nearly outdated, and has already been largely commented, the present report in its current form may be suboptimal.

My recommendation for the present report to strengthen its scientific message for students and non-expert readers in statistics would be to also provide a clear message on the added-value of Bayesian statistics, for example in an introduction that does not exist yet and that could be short but written in a didactic way. This would provide a theoretical context that may be beneficial to everyone.

6. PLOS authors have the option to publish the peer review history of their article (what does this mean?). If published, this will include your full peer review and any attached files.

Reviewer #1: No

Reviewer #2: No

---

## [Author Response · Author response to Decision Letter 0]

12 Nov 2020

Please see the attached document entitled "reviewer comments"

---

## [Editor Report · Decision Letter 1]

22 Dec 2020

A Bayesian reanalysis of the effects of hydroxychloroquine and azithromycin on viral carriage in patients with COVID-19

PONE-D-20-11807R1

Dear Dr. Hulme,

We’re pleased to inform you that your manuscript has been judged scientifically suitable for publication and will be formally accepted for publication once it meets all outstanding technical requirements.

Kind regards,

Jean Daunizeau

Academic Editor

PLOS ONE
---

## [Editor Report · Acceptance letter]

29 Jan 2021

PONE-D-20-11807R1 

A Bayesian reanalysis of the effects of hydroxychloroquine and azithromycin on viral carriage in patients with COVID-19 

Dear Dr. Hulme:

I'm pleased to inform you that your manuscript has been deemed suitable for publication in PLOS ONE. Congratulations! Your manuscript is now with our production department. 

Kind regards, 

on behalf of

Dr. Jean Daunizeau 

Academic Editor

PLOS ONE